# Effects of Cu, Zn Doping on the Structural, Electronic, and Optical Properties of α-Ga_2_O_3_: First-Principles Calculations

**DOI:** 10.3390/ma16155317

**Published:** 2023-07-28

**Authors:** Hui Zeng, Meng Wu, Meijuan Cheng, Qiubao Lin

**Affiliations:** 1College of Science, Hunan University of Science and Engineering, Yongzhou 425199, China; 2Fujian Provincial Key Laboratory of Semiconductors and Applications, Collaborative Innovation Center for Optoelectronic Semiconductors and Efficient Devices, Department of Physics, Xiamen University, Xiamen 361005, China; 3College of Science, Jimei University, Xiamen 361021, China

**Keywords:** α-Ga_2_O_3_, first-principles, doping, formation energies, electronic structure, optical properties

## Abstract

The intrinsic n-type conduction in Gallium oxides (Ga_2_O_3_) seriously hinders its potential optoelectronic applications. Pursuing p-type conductivity is of longstanding research interest for Ga_2_O_3_, where the Cu- and Zn-dopants serve as promising candidates in monoclinic β-Ga_2_O_3_. However, the theoretical band structure calculations of Cu- and Zn-doped in the allotrope α-Ga_2_O_3_ phase are rare, which is of focus in the present study based on first-principles density functional theory calculations with the Perdew–Burke–Ernzerhof functional under the generalized gradient approximation. Our results unfold the predominant Cu^1+^ and Zn^2+^ oxidation states as well as the type and locations of impurity bands that promote the p-type conductivity therein. Furthermore, the optical calculations of absorption coefficients demonstrate that foreign Cu and Zn dopants induce the migration of ultraviolet light to the visible–infrared region, which can be associated with the induced impurity 3d orbitals of Cu- and Zn-doped α-Ga_2_O_3_ near the Fermi level observed from electronic structure. Our work may provide theoretical guidance for designing p-type conductivity and innovative α-Ga_2_O_3_-based optoelectronic devices.

## 1. Introduction

Gallium oxides (Ga_2_O_3_) have attracted much attention for use in various applications such as solar-blind ultraviolet photodetectors [1,2], high-power transistors [3,4], Schottky diodes [5,6], as well as photocatalysts [7] due to its outstanding physical and chemical properties. Ga_2_O_3_ typically exhibits five different crystal phases, and the β phase is the most stable and intensively explored one [8,9]. However, other crystalline phases have been relatively sparsely studied; among them, α-Ga_2_O_3_ is the second stable phase. The metastable α-Ga_2_O_3_ phase is endowed with a corundum structure and belongs to the rhombohedral phase (R-3ch). Compared with β-Ga_2_O_3_, α-Ga_2_O_3_ possesses a wider bandgap (5.3 eV) [10], larger Baliga figure of merit (~3844) [11], and higher breakdown electrical field (8–10 MVcm^−1^) [12]. In recent years, α-Ga_2_O_3_ films and various heterojunctions have been successfully synthesized via different experimental devices, such as laser molecular beam epitaxy [13,14], chemical vapor deposition [15,16], halide vapor phase epitaxy [17], as well as various heterojunctions, such as on Al_2_O_3_ [18,19], ZnO [20], etc.

Likewise β-Ga_2_O_3_, perfect α-Ga_2_O_3_ shows n-type conductivity characteristics due to the inevitable introduction of native defects and impurities in the experiment, which seriously hinders its applications [21,22]. In order to construct p-n junctions for further applications of α-Ga_2_O_3_, it is imperative to develop and explore p-type conductivity. In general, doping technology can be an effective method to improve the conductivity, especially for a wide bandgap semiconductor [23,24,25,26,27,28,29,30]. Among different doping candidates, Cu and Zn are widely studied in β-Ga_2_O_3_ due to the p-type conductivity therein. Cu-doped β-Ga_2_O_3_ was found to be a promising p-type semiconductor due to the introduction of two acceptor impurity levels towards the top of the valence band researched via first-principles calculated methods [31]. Further electron paramagnetic resonance spectroscopy analyses denoted that Cu^2+^ preferentially sited on the octahedral coordination Ga site in the Ga_2_O_3_ lattice [32]. Li et al. illustrated that Zn-doped β-Ga_2_O_3_ generated a shallow energy state near the valence band maximum, which made a typical p-type Zn-doped β-Ga_2_O_3_ material [33]. The p-type conductivity was observed in β-Ga_2_O_3_ nanowires containing various amounts of Zn doping contents using the CVD method [34] and in Zn-doped β-Ga_2_O_3_ film fabricated using the pulsed laser deposition method [35]. In addition, about 1.0 μ_B_ magnetic moment was gained in the Zn-doped β-Ga_2_O_3_ supercell, which mainly originated from the O 2p orbitals near the doped Zn atom [36].

However, no further theoretical reports are available for Cu-/Zn-doped α-Ga_2_O_3_, to the best of our knowledge. Thus, systematic research studies on the electronic properties of Cu-/Zn-doped α-Ga_2_O_3_ are highly demanded. Research interests in the doping-dependent optical properties of α-Ga_2_O_3_ have been addressed recently. The systematic studies of 3d–5d transition metal doped α-Ga_2_O_3_ suggested that the induction of IB and IIB transition metal dopants can be endowed with low formation energies and could result in the optical absorption migration from deep ultraviolet to infrared [12]. Pan et al. denoted that added Cu, Ag, and Au elements in α-Ga_2_O_3_ led to the transformation from the ultraviolet to the visible light region [37].

Inspired by the doping-induced p-type conductivity in β-Ga_2_O_3_, two promising p-type dopants of Cu and Zn elements are studied. We performed first-principles calculations to investigate the structural, electronic, and optical properties of Cu- and Zn-doped α-Ga_2_O_3_. The detailed distributions of electron density, the defect formation energies and charge transitional levels under different crystal growth conditions, the electronic structure, as well as the optical properties are researched. This study is useful for understanding the utilization of Cu- and Zn-doped α-Ga_2_O_3_. We hope our work can provide theoretical guidance for designing α-Ga_2_O_3_-based functional materials as well as the promising applications of α-Ga_2_O_3_ for innovative optoelectronic devices.

## 2. Calculation Methods

### 2.1. Computational Details

In this work, our first-principles calculations adopt the Vienna ab initio Simulation Package (VASP) [38,39] using DFT [40] containing projected augmented wave (PAW) potentials. The generalized gradient approximation (GGA) parameterized by Perdew–Burke–Ernzerhof (PBE) [41] is employed to describe the interactions of exchange-correlation. The kinetic energy cutoff is set as 450 eV, the energy convergence criterion for the calculations is set to 1 × 10^−5^ eV/atom, and all the atomic locations have been fully tuned. When all residual forces are less than 0.01 eV/Å, the relaxation will be terminated. The valence electronic configurations of Ga, O, Cu, and Zn atoms are [Ar] 3d^10^4s^2^4p^1^, [He] 2s^2^2p^4^, [Ar] 3d^10^4s^1^, and [Ar] 3d^10^4s^2^, respectively.

In this study, a 2 × 2 × 1 α-Ga_2_O_3_ supercell is modeled containing 48 Ga atoms and 72 O atoms, in which one Cu or Zn impurity replaces the Ga position with an equivalent doping concentration of 2.08 %, as shown in Figure 1. A 4 × 4× 2 Monkhost-Pack grid is used for the structural relaxation, while a 9 × 9 × 4 Monkhost-Pack grid is employed for the calculations of the density of states (DOS) and optical properties. The tetrahedron method is adopted to give a good account of the DOS calculations.

### 2.2. Formation Energy, Transitional Level, and Optical Calculations

For defect D doping in α-Ga_2_O_3_, the formation energy in the charge state q is determined as [21,42]
(1)$$H_{D,q}(E_{f},\mu )=[E_{D,q}-E_{p}]+\sum_{i}n_{i}\mu_{i}+q(E_{VBM}+E_{f})+E_{corr}$$
where $E_{D,q}$ and $E_{p}$ represent the total energy of the defect and perfect supercell, respectively. $n_{i}$ denotes the number of i atoms added ($n_{i}<0$) or removed ($n_{i}>0$) from the perfect supercell, and $\mu_{i}$ is the corresponding chemical potential of the impurity or host atom. $E_{VBM}$ is the energy of valence band maximum (VBM). $E_{f}$ is the Fermi level, which is referenced to the VBM in the bulk. The value of $E_{f}$ is set to zero at VBM and can range from 0 to the energy value of conduction band minimum (CBM). $E_{corr}$ is associated with finite-size corrections, is determined by the potential alignment, and is given as [42]
(2)$$E_{corr}=q(V_{D,q}^{r}-V_{p}^{r})$$

Here, the potential difference between the charged defect Ga_2_O_3_ supercell ($V_{D,q}^{r}$) and perfect Ga_2_O_3_ supercell ($V_{p}^{r}$) are calculated from the atomic sphere-averaged electrostatic potentials at the atomic sites farther away from the defect, which is calculated via the software of VASPKIT Standard Edition 1.3.5 [43].

Note that the chemical potential satisfies the boundary conditions as follows:(3)$$2\mu_{Ga}+3\mu_{O}=\mu_{Ga_{2}O_{3}},\;\;\;\;\;\;\;\;\mu_{Ga}\leq \mu_{Ga}^{Metal},\;\;\;\;\;\;\;\;\;\;\;\mu_{O}\leq \frac{1}{2}\mu_{O_{2}}$$

In terms of the different synthesis conditions for gallium oxide, it can be divided into two categories, Ga-rich and O-rich, for the calculations of chemical potential. Under O-rich growth condition,
(4)$$\mu_{O}=\frac{1}{2}\mu_{O_{2}},\;\;\;\;\;\;\;\;\;\;\;\;\;\;\;\mu_{\mathrm{Ga}}=\frac{1}{2}(\mu_{Ga_{2}O_{3}}-\frac{3}{2}\mu_{O_{2}})$$

Under Ga-rich growth condition,
(5)$$\mu_{Ga}=\mu_{Ga}^{Metal},\;\;\;\;\;\;\;\;\;\;\;\;\;\;\;\mu_{O}=\frac{1}{3}(\mu_{Ga_{2}O_{3}}-2\mu_{Ga})$$
where $\mu_{Ga_{2}O_{3}}$ is the chemical potential of bulk β-Ga_2_O_3_. The chemical potential of $\mu_{Ga}^{Metal}$, $\mu_{\mathrm{Cu}}$ and $\mu_{\mathrm{Zn}}$ are calculated from the energies of the most stable bulk crystal of the Ga, Cu, and Zn atoms, respectively, while $\mu_{O}$ is gained from the energy of O_2_. The chemical potentials of $\mu_{O}$, $\mu_{Ga}^{Metal}$, $\mu_{\mathrm{Cu}}$, and $\mu_{\mathrm{Zn}}$ are −4.92 eV, −7.48 eV, −3.72 eV, and −1.11 eV, respectively, under O-rich condition, while the values are −7.96 eV, −2.90 eV, −3.72 eV, and −1.11 eV, respectively, for Ga-rich (oxygen-deficient) environment.

The transition energy $\varepsilon (q_{1}/q_{2})$ between charge state q_1_ and q_2_ for defect D doping configuration is calculated as [44]
(6)$$\varepsilon (q_{1}/q_{2})=\frac{E_{D}^{q_{1}}{|}_{E_{f}=0}-E_{D}^{q_{2}}{|}_{E_{f}=0}}{q_{2}-q_{1}}$$
where $E_{D}^{q}{|}_{E_{f}=0}$ represents the formation energy of the defect D in charge state q evaluated at $E_{f}=0$. The $\varepsilon (q_{1}/q_{2})$ denotes the E_f_ position where the charge state q_1_ and q_2_ have equal formation energy.

The absorption coefficients in optical properties are described as follows [45,46]:(7)$$\alpha (\omega )=\sqrt{2}\omega {\left[\sqrt{\varepsilon_{1}^{2}(\omega )+\varepsilon_{2}^{2}(\omega )}-\varepsilon_{1}(\omega )\right]}^{\;1/2}$$
where $\varepsilon_{1}(\omega )$ and $\varepsilon_{2}(\omega )$ indicate the real and imaginary part of the dielectric function, respectively. The $\varepsilon_{2}(\omega )$ can be obtained using the following equation:(8)$$\varepsilon_{2}(\omega )=\left(\frac{4\pi^{2}e^{2}}{m\omega^{2}}\right){\sum_{i,j}\int \langle i\left|M\left|j\right.\right.\rangle }^{2}f_{i}(1-f_{i})\times \delta (E_{jk}-E_{ik}-\omega )d^{3}k$$

Here, m, M, e, and ω denote the mass of free electrons, the dipole matrix, the electron charge, and the frequency of incident photons, respectively. i, j, $f_{i}$, and k represent the initial state, final state, Fermi distribution function, and wave function vector, respectively, while the $\varepsilon_{1}(\omega )$ is calculated by the equation
(9)$$\varepsilon_{1}(\omega )=1+\frac{2}{\pi }P\int_{0}^{\infty }\frac{\omega^{\prime }\varepsilon_{2}(\omega^{\prime })d\omega^{\prime }}{{\omega^{\prime }}^{2}-\omega^{2}}$$
where P represents the principle value of the integral. In addition, the $\varepsilon_{2}(\omega )$ is related to the absorption of light and dielectric loss of energy, while $\varepsilon_{1}(\omega )$ is associated with the stored energy.

## 3. Results and Discussions

### 3.1. Structural Stability

The calculated lattice parameters of perfect α-Ga_2_O_3_ are a = b = 5.055 Å and c = 13.586 Å, which are in good agreement with the literature values for bulk α-Ga_2_O_3_ [37,47,48], as shown in Table 1. The lattice constants remain almost unchanged for the Cu- and Zn-doped α-Ga_2_O_3_ supercell, which can be attributed to the fact that the Cu and Ga atoms have identical ionic radii and local structure, as is likewise for Zn and Ga atoms. The variation in the radii between Cu^1+^ (Cu^2+^) and Ga^3+^ ions is 24.2% (17.7%), while it is 19.4% for Zn^2+^ and Ga^3+^ ions.

The distribution of electron density is employed to evaluate the crystal bonding characteristic. Figure 2a shows the electron density of perfect α-Ga_2_O_3_; the electrons around Ga and O atoms illustrate a strong covalent bonding between Ga and the nearest neighbor O atoms. For the case of Cu doping, as shown in Figure 2b, the arrangement of the atoms has a minor alteration, which is consistent with the variation of lattice constants as discussed above. The dispersed electrons of the Cu atom in the backdrop can be attributed to the minimal covalent bonding effect. Meanwhile, the decreased electron density of the O atom adjacent to the doped Cu atom indicates that a small number of electrons migrate from O atoms to the nearby Cu atoms, as revealed by the electron density analysis of the O atoms; thus, Cu-doped α-Ga_2_O_3_ can be a possible p-type doping. The electron density distribution in the Zn-doped case is shown in Figure 2c, which has similar features as those of Cu doping.

To further study the structural stability of the Cu- and Zn-doped α-Ga_2_O_3_ supercell, the defect formation energies under different conditions are calculated, as shown in Figure 3. Meanwhile, in order to assess the ionization energies and effectiveness of doping in α-Ga_2_O_3_ systems, we employ the transition levels. The formation energies of Cu- and Zn-doped α-Ga_2_O_3_ under an O-rich atmosphere are shown in Figure 3a, and the dashed line represents the calculated band gap of α-Ga_2_O_3_. Our calculated value of band gap for perfect α-Ga_2_O_3_ is 2.50 eV. We note that the characteristics of band orbital states are consistent with those obtained from previous studies, but the band gap value is smaller than the experimental value, as shown in Figure 1b and Figure 4a [45,49]. The underestimated band gap for DFT calculation is common, but it has no effect on our conclusions qualitatively [50,51].

The formation energies for the Cu and Zn doping cases under O-rich conditions are shown in Figure 3a, which possess negative values throughout the band gap, indicating that both elements can easily be doped in α-Ga_2_O_3_. This can be attributed to the fact that the three elements (Cu, Zn, and Ga) are next to each other in the periodic table of elements, and thus the ionic radii between dopants (Cu, Zn) and host Ga are comparable, as discussed above.

The defect concentration can be stated as follows [49,52]:(10)$$c=N_{site}\exp(-H/K_{B}T)$$
where c, $N_{site}$, H, $K_{B}$, and T denote the effective doping concentration, the number of doping, the formation energy and the Boltzmann constant and temperature, respectively. According to the equation, the lower formation energy in the Zn-doped α-Ga_2_O_3_ system corresponds to a higher effective doping concentration compared with that of Cu-doped α-Ga_2_O_3_.

Under O-rich conditions, Figure 3a illustrates that the positively charged and negatively charged Cu are energetically favorable when the Fermi level approaches the VBM and CBM, respectively, whereas the negatively charged Zn are dominant across the entire band gap. The transition level ε(+1/0) of Cu-doped α-Ga_2_O_3_ is situated at 0.61 eV, which is far above the VBM and acts as a deep acceptor energy level. In addition, the transition levels ε(+2/+1), ε(0/−1), and ε(−1/−2) are 0.26, 1.01, and 2.03 eV, respectively measured from the VBM. For the Zn-doped case, the +2 charge state is observed in a limited region around the VBM, as shown in the inserted figure in Figure 3a. The transition level (+2/0) occurs at 0.02 eV above the VBM; thus, a shallow acceptor level is expected for Zn-doped α-Ga_2_O_3_. The transition level ε(0/−1) is 0.06 eV and ε(−1/−2) is 2.52 eV, which is beyond the CBM. It is worth mentioning that during the growth of α-Ga_2_O_3_, some native defects such as Ga_i_ and V_O_ are unintentionally introduced and give rise to the n-type conduction characteristic [21]. As a result, the Fermi level always tends to be positioned in the region of high α-Ga_2_O_3_ bandgap. Therefore, the −2 and −1 charge states, i.e., Cu^1+^ and Zn^2+^ oxidation states, are the predominant states for the Cu- and Zn-doped α-Ga_2_O_3_ supercell, respectively. In the meantime, the −1 charge states (Cu^2+^) are an alternative option because the location is near the CBM. Accompanying the valence state variations, two-hole and one-hole introductions for Cu- and Zn-doped α-Ga_2_O_3_ are expected, respectively, corresponding to double deep acceptor levels for Cu and one shallow acceptor level for Zn. As a result, the Zn atom can significantly improve the carrier concentration and is likely to be the effective hole dopant in α-Ga_2_O_3_, while the Cu atom can effectively compensate electrons in native donor-type defects and can significantly change the non-equilibrium carrier lifetime, considering the shallow doping for the Zn atom and deep doping for the Cu atom. For the Ga-rich condition, i.e., O-poor environment, as shown in Figure 3b, the tendency is the same as in an O-rich atmosphere with the exception of higher formation energies. Therefore, Cu and Zn impurities are more easily substituted to Ga sites under O-rich conditions.

### 3.2. Electronic Structure

In order to explore the orbital contribution of impurity atoms, the calculated total density of states (TDOS) and partial density of states (PDOS) for perfect, Cu-, and Zn-doped α-Ga_2_O_3_ are shown in Figure 4. Figure 4a illustrates that the VBM of perfect α-Ga_2_O_3_ is predominantly composed of O 2p orbital-derived states with minor hybridization with Ga 3d and 4p orbitals, while the CBM is composed mainly of Ga 4s orbitals [49]. Additionally, the strong coupling of atomic orbital interaction between Ga and O atoms implies that Ga-O bonds have a covalent bond feature, which is in accordance with the results of electron density distribution.

For the Cu dopant, as shown in Figure 4b, the induced impurity levels are mainly composed of the Cu 3d orbitals near the Fermi level, and it is not fully occupied. The 3d states of the Cu dopant are hybridized obviously with the newly generated occupied O 2p orbitals and tiny Ga 3d orbitals near the Fermi level, implying a strong exchange interaction among them and the formation of a covalent Cu-O bond. In addition, the hole doping can decrease the Fermi level, as shown in Figure 4b. For the Zn doping case in Figure 4c, the results are very similar to that of Cu-doped α-Ga_2_O_3_ except for a relatively shallow acceptor level (approximately 0), which matches with the data of formation energy.

### 3.3. Optical Property

Impurity levels induced by a dopant can affect the characteristic of electronic properties, which can further influence the optical absorption of the material [53]. Figure 5a shows the optical absorption coefficients of perfect, Cu-doped, and Zn-doped α-Ga_2_O_3_ vary in energy from 0 to 30 eV, respectively. The strong absorption peak at 12.5 eV for perfect α-Ga_2_O_3_ suggests the ultraviolet properties, which can be associated with the band migration from the O 2p occupied orbitals to the Ga 4s unoccupied orbitals. One can notice that perfect α-Ga_2_O_3_ is endowed with strong and weak optical absorption in the ultraviolet and visible–infrared region, respectively, because of its wide band gap. The profiles of Cu- and Zn-doped α-Ga_2_O_3_ in the high energy ultraviolet region observed from the insert of Figure 5a are similar to that of perfect α-Ga_2_O_3_, except for slightly lower absorption peaks at 9.7 eV and 12.5 eV. It indicates that the two foreign dopants slightly weaken the optical absorption coefficients for α-Ga_2_O_3_ in the ultraviolet region. Importantly, new small peaks are created in the lower region. Figure 5b shows the detailed diagram of energy change from 0 to 5 eV. The perfect α-Ga_2_O_3_ possesses an optical band gap of about 2.5 eV, which is in good agreement with the results of the electronic structure. When introducing the Cu and Zn dopants into α-Ga_2_O_3_, the absorption coefficients of new peaks are relatively low but result in the transformation from the ultraviolet light region to the visible–infrared region (considering the underestimated band gap). For Cu- and Zn-doped α-Ga_2_O_3_, the 3d orbitals dominate these impurity levels near the Fermi level, which can usually enhance the absorption coefficients in the infrared or visible region. As analyzed above, double deep acceptor levels for Cu and one shallow acceptor level for Zn are expected; therefore, two absorption peaks and one absorption peak are present for Cu and Zn, respectively, as shown in Figure 5b. The main peak at 1.53 eV and 0.16 eV for Cu-doped α-Ga_2_O_3_ can be associated with the transition from O 2p orbitals to Cu 3d orbitals and inter-band transition between the two induced holes, respectively. The main peak at 0.48 eV for Zn-doped α-Ga_2_O_3_ can be related to the transition from O 2p orbitals to Zn 3d orbitals.

### 3.4. Conclusions

The detailed distributions of electron density, defect formation energies, and charge transitional levels under different crystal growth conditions, as well as the electronic and optical properties for Cu- and Zn-doped α-Ga_2_O_3_, are discussed based on first-principles DFT calculations with the GGA method. The distribution of electron density illustrates that a small number of electrons transfer to the doping atom. However, double deep acceptor levels for Cu and one shallow acceptor level for Zn are expected. Thus, the Zn atom can significantly improve carrier concentration and is believed to be the effective hole dopant in α-Ga_2_O_3_, while the Cu atom can compensate electrons in native defects and significantly change the non-equilibrium carrier lifetime. The 3d states of the Cu and Zn dopants are obviously hybridized with the newly generated occupied O 2p orbitals and tiny Ga 3d orbitals near the Fermi level, which forms the covalent Cu-O and Zn-O bonds. When introducing the Cu and Zn dopants into α-Ga_2_O_3_, the absorption coefficients of new peaks are relatively low but result in the optical absorption migration from deep ultraviolet light to visible–infrared light. The main peak of optical absorption at 1.53 and 0.16 eV for Cu-doped α-Ga_2_O_3_ can be associated with the transition from O 2p orbitals to Cu 3d orbitals and inter-band transition between the two induced holes, respectively. The main peak of optical absorption at 0.48 eV for Zn-doped α-Ga_2_O_3_ can be related to the transition from O 2p orbitals to Zn 3d orbitals.

## Figures and Tables

**Figure 1 materials-16-05317-f001:**
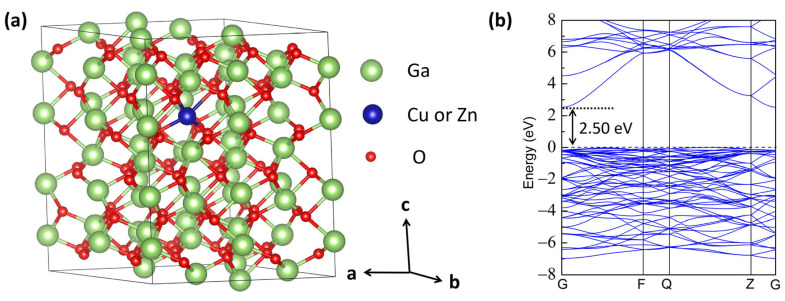
(**a**) The model of Cu and Zn-doped α-Ga_2_O_3_ supercell obtained from VESTA. The larger green spheres and the smaller red spheres represent Ga and O atoms, respectively, while the blue sphere denotes the substituted Ga doping site with one Cu or Zn dopant. The a, b, and c axes denote the crystallographic *a*, *b*, and *c* directions, respectively. (The reader is suggested to see the web version of this article for interpretation of the color references in this figure legend.) (**b**) The band structure for perfect α-Ga_2_O_3_ supercell.

**Figure 2 materials-16-05317-f002:**
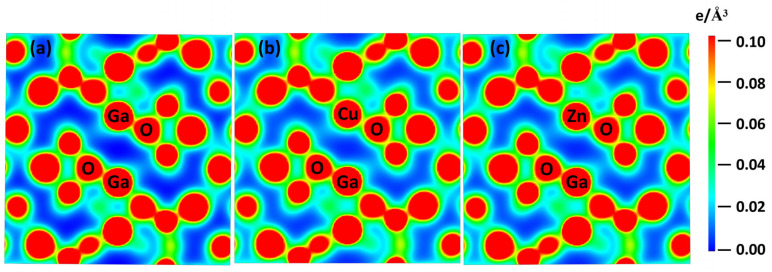
The electron density distribution for (**a**) perfect α-Ga_2_O_3_, (**b**) Cu-dope α-Ga_2_O_3_, and (**c**) Zn-doped α-Ga_2_O_3_. The red and blue colors represent higher and lower electron density, respectively.

**Figure 3 materials-16-05317-f003:**
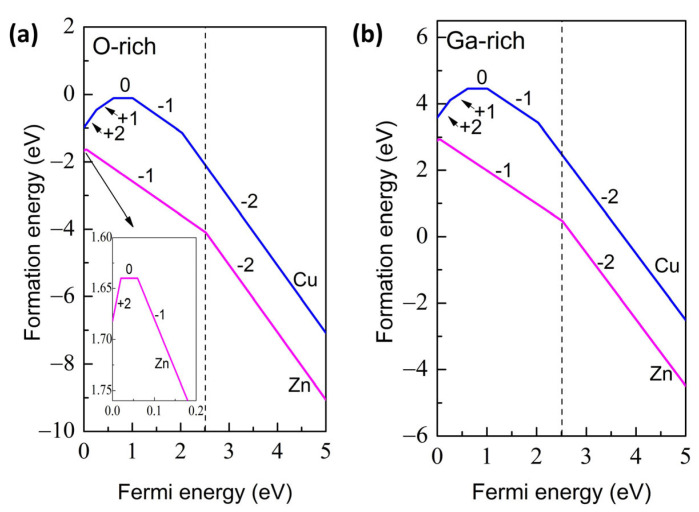
The defect formation energies of Cu- and Zn-doped α-Ga_2_O_3_ under (**a**) the O-rich and (**b**) Ga-rich conditions. The dashed line represents the band gap of perfect α-Ga_2_O_3_.

**Figure 4 materials-16-05317-f004:**
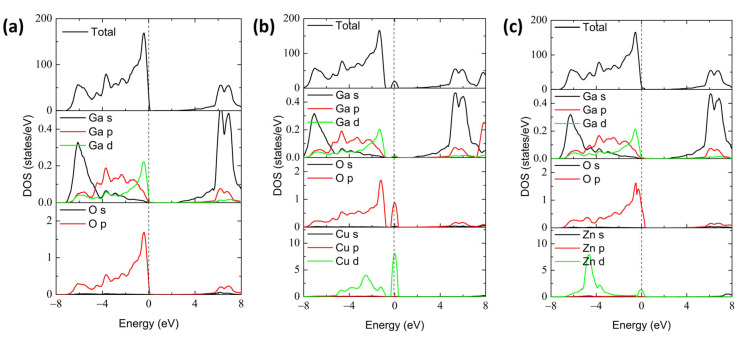
Calculated total density of states (TDOS) and partial density of states (PDOS) of (**a**) perfect α-Ga_2_O_3_, (**b**) Cu-doped α-Ga_2_O_3_, and (**c**) Zn-doped α-Ga_2_O_3_. Dashed line denotes the Fermi level.

**Figure 5 materials-16-05317-f005:**
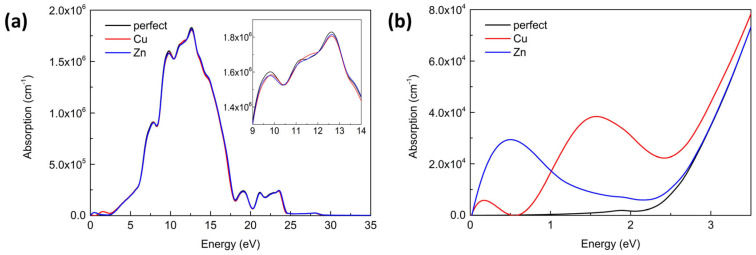
The optical absorption of perfect, Cu-doped, and Zn-doped α-Ga_2_O_3_ in energy change from (**a**) 0 to 30 eV and (**b**) 0 to 5 eV.

**Table 1 materials-16-05317-t001:** The calculated lattice constants for perfect, Cu-doped, and Zn-doped α-Ga_2_O_3_.

Lattice Constants	Perfect (This Work)	Perfect (Literature)	Cu-Doped	Zn-Doped
a (Å)	5.055	5.06 [47]/5.076 [37]/4.983 [48]	5.056	5.059
c (Å)	13.586	13.63 [47]/13.703 [37]/13.433 [48]	13.574	13.583
V (Å^3^)	1202.636	1222.8 [37]	1202.029	1204.146

## Data Availability

No new data were created or analyzed in this study. Data sharing is not applicable to this article.

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
