# Peer review of "Effects of Cu, Zn Doping on the Structural, Electronic, and Optical Properties of α-Ga2O3: First-Principles Calculations"

_materials, 2023, doi:10.3390/ma16155317_

Round 1

Reviewer 1 Report

In this study, the first principal calculations were performed to understand the effect of Cu , Zn doping on the structural, optical, and electronic properties of Gallium oxide. The topic is interesting and has optoelectronic. The abstract and introduction are fairly explained. Please introduce a paragraph in the introduction that emphasizes the novelty of the present work. Please highlight the significance of the study.

How the structure in Figure 1 (a) was obtained? Is it through VESTA or , material studio? Please mention it

Figure 5 is not clear, please improve the size and change the legend to line or dotted to clear the visibility.

Is there any role of temperature on the properties if yes what temperature was chosen for the calculations?

How validation is performed?

What was the doping concentration? and how the concentration has affected the properties.

I can see it is mentioned that optoelectronic properties are mentioned how other properties are claimed to justify? Please explain

Please improve the quality of the Figures.

Reviewer 2 Report

In this study, two promising p-type dopants of Cu and Zn elements are studied. I think, that paper is interesting, I would propose some changes as follows:

1.      The Authors should clearly stress the message of the manuscript in the abstract.

2.      It would be desirable to confirm the obtained results by performing the experiments.

3.      The Authors should justify the chosen range of the Fermi energy. Is 5eV value the real case?

4.      The paper requires proofreading.

5.      Authors are missing some recent articles in the field describing the optical properties such as Tunable terahertz structure based on graphene hyperbolic metamaterials, etc.

The paper requires proofreading.

Reviewer 3 Report

The authors approach the effects of Cu, Zn doping on the structural, electronic and optical properties of α-Ga2O3 by first-principles calculations. Modeling details and choices are highly adequate and well-chosen, especially having in mind the materials system studied, and the properties examined. Thus, the results look credible besides being very well presented. Equally important, the discussion is feasible and fits the current research questions regarding doping effects on wide range of properties of α-Ga2O3.

All in all, this work represents a valuable contribution with possible wider impact in the field of α-Ga2O3.

The authors chose an adequate structure of the manuscript. Concise, and nicely illustrated figures and their corresponding analysis are provided.

There are some issues with this already excellent manuscript that will need to be addressed before the manuscript becoming suitable for publication, i.e., it can be considered for publication after a minor revision:

1: Title: The word “from” maybe omitted from the title and substituted by colon “:”.

2: Abstract should briefly mention the level of theory employed (PBE, in this case).

3: Have the authors studied (or are they aware of work by other teams concerning) any thermal aspects of the Cu, Zn doping, for example by molecular dynamcs?

4: In the introduction, the authors should also mention that aspects of similar semiconductor systems have already been studied by using theoretical methodology at the same level of theory, namely [CrystEngComm 23 (2021) 6661-6667; Physical Review B 68 (2003) 241401; ACS Nanoscience Au 3 (2022) 84-93].

5: Spell-check and stylistic revision of the paper are necessary. Some long sentences, as well as misspellings, etc., are noticeable throughout the text.

Spell-check and stylistic revision of the paper are necessary. Some long sentences, as well as misspellings, etc., are noticeable throughout the text.
